

# The effect of different initial solutions on the metaheuristic algorithms for the single allocation p-hub center and routing problem

Abdul Kader Kassoumeh[1], Zühal Kartal[2] and Ahmet Arslan[1]

[1] Department of Computer Science, Eskisehir Technical University, Eskisehir, Turkiye
[2] Department of Industrial Engineering, Eskisehir Technical University, Eskisehir, Turkiye

## ABSTRACT

This article introduces methods for initializing a single-trajectory-based metaheuristic, specifically a simulated annealing (SA) algorithm, using constructive heuristics. These methods are designed to target promising regions within the search space of an nondeterministic polynomial time (NP)-hard problem, namely the single allocation p-hub center and routing problem. The objective of this problem is to allocate demand centers to hubs and design vehicle routes such that the maximum distance between all origin-destination pairs is minimized. To analyze the impact of different initial solutions, various constructive heuristics, including greedy and hybrid strategies, have been proposed. Additionally, a problem decomposition approach leveraging domain-specific knowledge has been incorporated through a matheuristic initial solution strategy to enhance the efficiency of the simulated annealing algorithm. This approach generates high-quality initial solutions by first solving the p-hub center problem and then using the obtained hubs and their assignments as inputs to the min-max multiple traveling salesman problem. In this problem, the objective function is formulated differently from the literature by minimizing the longest distance between the two nodes. Several experiments have been conducted on the Turkish network, and upon examining the results, it has been observed that each initial solution generation strategy provides improvements in problem instances with specific characteristics, such as the number of vehicles and nodes. We also observed lower objective function values for all medium- and large-sized test problems taken from the literature, highlighting the effectiveness of the proposed strategies.

# INTRODUCTION

Heuristic algorithms are methods developed to solve problems more efficiently when traditional techniques are either too slow for finding an exact or approximate solution or unable to find any exact solution within a given search space. The goal of a heuristic algorithm is to generate a solution within a reasonable time frame that is sufficiently effective for addressing the problem at hand. Although this solution may not be the

Corresponding author
Zühal Kartal,
zkartal@eskisehir.edu.tr

optimal one or might only approximate the exact solution, it remains valuable because it can be obtained without taking an excessive amount of time.

Generally, heuristics are classified into improvement and constructive heuristics, based on their respective strategies to solve the problems. Constructive heuristics begin their approach with an empty solution, subsequently employing an iterative process to progressively build the complete solution. Hence, constructive heuristics do not explore the entire solution space. Instead, they utilize a specific type of strategy to determine the approach for constructing the whole solution. As implied by their name, constructive heuristics generate solutions by composing their individual components, as opposed to improving whole solutions, which is accomplished by sequentially introducing one element at a time to an incomplete (partial) solution. Constructive heuristics commonly adjust and utilize greedy algorithms, where the best possible element is incorporated during each step. Nonetheless, due to using some greedy approach in building whole solutions, constructive heuristics might have the potential to obtain local optima.

Improvement heuristics require an initial solution to start the exploration of the solution space for the given problem, aiming to reach the best possible solution. This is done through a series of iterative steps, each deploying a range of diverse neighborhood operations. Improvement heuristics have the potential to yield good solutions for combinatorial optimization problems due to their comprehensive exploration of the solution space.

Improvement heuristics rely on an initial solution to start the exploration of the solution space for the given problem, aiming to reach the best possible solution. There are two main sources of the initial solutions which are: a random solution generation mechanism and constructive heuristics. The former generates solutions using random processes, the latter is an iterative method. Both can quickly generate/build complete solutions from scratch that can be integrated into complex constructive or iterative heuristics.

It should be noted that constructive heuristics are designed with the problem at hand in mind. For instance, the "Savings Algorithm" of *Clarke & Wright (1964)* can be an example of such constructive heuristic when the particular problem to be addressed is the vehicle routing problem (VRP). Throughout the years, several variations and specializations of the VRP have been introduced (see *Elatar, Abouelmehdi & Riffi, 2023* for a comprehensive list of problem variants). Consequently, the Savings algorithm was reformulated to handle the time-window variant in *Solomon (1987)*, and it may well be extended to other VRP variants in the future. However, it cannot be used to solve an arbitrary combinatorial optimization problem such as scheduling. Thus, the problem and the heuristics are tightly coupled. For this very reason, the existing studies investigating the impact of the initial solution, in which problems and heuristics to form initial solutions are closely intertwined, consider only a single problem type per publication at a time.

Metaheuristic algorithms, in contrast to heuristic algorithms which are designed to solve a wide range of optimization problem types, are adaptable and can be employed across various problem domains. They are generally classified into two groups: population-based or single trajectory. The former concentrate on maintaining and enhancing multiple candidate solutions (genetic algorithms, ant colony *etc.*), while the

latter focus on modifying and improving a single initial solution (simulated annealing, tabu search *etc.*). The initial solution in a metaheuristic algorithm constitutes a critical element impacting its capacity to ascertain solutions of superior or acceptable quality. In the absence of any prior domain-specific information about the problem at hand, it is customary for the initial solution to be predominantly formed through entirely random procedures. When information about the problem is available, constructive heuristics can be utilized to generate an initial solution.

Metaheuristic algorithms typically comprise iterative optimization algorithms, often sharing a common step in their algorithmic framework, namely solution(s) initialization. Within the research community, it is widely acknowledged that this initialization step holds significant importance in the optimization process. This is because all subsequent solutions generated rely, to some extent, on their predecessors and ultimately on the initial solution or initial population of solutions. In the literature, the impact of initial solutions on metaheuristics has predominantly been examined within the context of population-based metaheuristics. As pointed out in a recent by review article by *Sarhani, Voß & Jovanovic (2023)*, which clearly reveals the research gap on initialization of metaheuristics, including single trajectory-based metaheuristics.

The 'initial solution effects' line of research is summarized in the following paragraph. These problems extend from the quadratic assignment problem to energy resource planning, and from the production routing problem to transmission expansion planning. The effects of the initial solution on an algorithm are examined as stated above, and each article addresses a specific problem.

In *Liu*'s *(2021)* study, the impact of the initial solution was examined for a genetic algorithm developed for the *quadratic assignment problem*. The following initial solution strategies were proposed: random, random plus local search, random plus bad local search, a greedy randomized adaptive search procedure (GRASP), and a greedy randomized adaptive search procedure plus local search. When the genetic algorithm ran on top of different initial solution strategies, the authors observed that the quality of the final (best-found) solution was formed in reverse order to the above ranking. *Antonio, Ramon & Ruben (2011)* investigated how selecting good initial populations for genetic algorithms influences both the speed of convergence and the overall quality of the final solutions of transmission expansion planning problem. *Sousa et al. (2016)* proposed five different initial solutions for a simulated annealing algorithm to solve the energy resource scheduling in smart grids. These initial solutions are (a) random solution; (b) ant colony optimization; (c) naive-scheduling heuristic; (d) pre-scheduling heuristic; and (e) mixed-integer linear programming. The authors used the results of a relaxed version by removing certain constraints from the resource scheduling problem in the mixed-integer linear programming based initial solution. *Ahmed, Hvattum & Agra (2023)* investigated the impact of the initial solutions obtained by mathematical modeling relaxation of a two-commodity flow formulation, a three-index flow formulation, and a four-index flow formulation for the production routing problem on the performance of the proposed matheuristic. *Oman & Cunningham (2001)* investigated how seeding genetic algorithms with high-quality initial solutions-evaluated based on their objective values-impacted

performance on the traveling salesman problem (TSP) and the job-shop scheduling problem. *Burke, Newall & Weare (1998)* applied various initialization strategies to the population of a genetic algorithm for the timetabling problem. In their study, *Fischetti, Ljubic & Sinnl (2016)* evaluated the performance of three distinct initial solutions for the prepack optimization problem using an *ad-hoc* heuristic. These initial solutions were constructed through three different approaches: random generation, selecting the most dissimilar options, and the most similar ones. Their effectiveness was further assessed by incorporating a refinement heuristic.

In line with the literature review provided in the above paragraph, this study also focuses on a single problem, namely the single allocation p-hub center and routing problem. The primary motivation of this study is to investigate the effects of three different constructive heuristics developed for the single allocation p-hub center and routing problem on a single trajectory metaheuristic, which is simulated annealing (SA). The reason for selecting SA is the need for a well-balanced interplay between diversification and intensification throughout the search process (*Assad & Deep, 2018*). Intensification, commonly referred to as exploitation, signifies the algorithm's ability to exploit the search space surrounding the current good solution, while diversification, also known as exploration, entails exploring new regions of the search space to introduce novel information. We also note here that SA had been selected due to its recognized capability to strike a proper balance between these two conflicting characteristics in combinatorial optimization problems and especially in hub location and routing problems (*Ernst & Krishnamoorthy, 1996*; *Kartal, Krishnamoorthy & Ernst, 2019*; *Ghaffarinasab, 2018*).

Despite the existence of a wide variety of metaheuristics in the literature, several interesting metaheuristic algorithms have also been introduced in recent years. Interested readers may examine a novel genetic algorithm for solving the fuzzy dynamic ship routing and scheduling problem (*Das et al., 2022*), a novel discrete rat swarm optimization algorithm designed for the quadratic assignment problem (*Mzili et al., 2023b*), a discrete penguins search algorithm proposed for the multiple traveling salesman problem (*Mzili, Mzili & Riffi, 2023*), and hybrid genetic and penguin search optimization algorithms developed for the flow shop scheduling problem (*Mzili et al., 2023a*).

In the literature, the performance of metaheuristic algorithms is influenced by factors such as parameter optimization and the quality of the initial solution used. While considerable attention has been devoted to parameter tuning, the impact of a strong initial solution has often been overlooked. Yet, an effective initialization can play a role similar to that of parameter tuning by increasing the likelihood of reaching better final objective function values, especially when early-stage diversification is incorporated. Most studies that focus on initialization tend to address population-based algorithms (*Li, Liu & Yang, 2020*; *Oman & Cunningham, 2001*), leaving its effect on single trajectory-based metaheuristics less explored.

Research focusing on the impact of initial solutions in single trajectory-based algorithms remains scarce. For instance, *Alfonzetti, Dilettos & Salerno (2006)* demonstrated that in SA, the algorithm restarts with a new starting solution when no improvement is observed; a mechanism that often involves randomly selecting a new starting point or choosing the

best candidate from a randomly generated pool to escape local optima. Similarly, *Li, Liu & Yang (2020)* showed that even if the initial solutions are not near optimal, the algorithm can still yield good outcomes provided that high diversity is maintained, and sufficient iterations are performed. This extensive search allows the algorithm to effectively explore the solution space, mitigating any drawbacks of an initial starting point.

Building on these insights, our study addresses this gap by investigating how the quality of the initial solution affects the final outcome in single trajectory-based metaheuristic algorithms. Specifically, we focus on the uncapacitated single allocation p-hub center and routing problem; a decomposable problem and evaluate three constructive heuristics, designed to exploit its structure within a simulated annealing framework.

The uncapacitated single allocation p-hub center and routing problem allocates demand centers to hubs and designs vehicle routes in such a way that the maximum distance is minimized between all origin-destination pairs (*Kartal, Krishnamoorthy & Ernst, 2019*). This problem is a combination of two well-known problems in the literature, namely the uncapacitated single allocation p-hub center (USApHCP) and the (multiple) traveling salesman problem (mTSP). USApHCP is NP-hard, which has been proven by *Ernst et al. (2009)*. The NP-Complete class of the traveling salesman problem (TSP) has been proven by *Karp (1972)*. Therefore, developing algorithms that are capable of obtaining optimal solutions for such problems within a reasonable time is not possible. As a result, a diverse range of heuristic algorithms can be employed to address both hub location and routing type problems. Although almost every hub location and routing problem in the literature takes into account different routing constraints and objective functions, in the following paragraph and Table 1 we provide an overview of both the metaheuristic and matheuristic algorithms developed for hub location and routing problems also the initial solutions strategies used strategies employed by these methods.

*De Freitas et al. (2023)* used a biased random-key genetic algorithm for the hub location routing problem with directed tours, incorporating a random initial solution. *Aloullal, Saldanha-da-Gama & Todosijević (2023)* introduced an input into matheuristic algorithms by relaxing integrality constraints formed by constraints involving only related to hub locations for the multi-period single-allocation hub location-routing problem, aiming to find a feasible solution. *Wang, Liu & Yang (2023)* developed a genetic algorithm for the robust hub location and routing problem with a third-party logistics strategy, generating initial solution populations based on random and greedy heuristics. For the multiple allocation hub location and routing problem, *Wu, Qureshi & Yamada (2022)* used a greedy approach as the initial solution for the adaptive large neighborhood search (ALNS) algorithm they developed. This approach was based on a two-stage construction phase. *Bütün, Petrovic & Muyldermans (2021)* proposed a tabu search algorithm for the capacitated directed cycle hub location and routing problem under congestion, suggesting a three-stage decomposition-based greedy heuristic for the initial solution. *Yang, Bostel & Dejax (2019)* developed a memetic algorithm for a hub location and routing problem with distinct collection and delivery tours, deriving an initial population consisting of random and greedy-based solutions. *Danach, Gelareh & Neamatian Monemi (2019)* solved the capacitated single-allocation p-hub location and routing problem using a feasible initial

**Table 1 Metaheuristics and initial solution strategies in hub location and routing related studies.**

| | Metaheuristics | Problem type | Initial solution strategy |
|---|---|---|---|
| *De Freitas et al. (2023)* | Biased random-key genetic algorithm | Hub location routing problem with directed tours | Random |
| *Aloullal, Saldanha-da-Gama & Todosijević (2023)* | Matheuristic | Multi-period single-allocation hub location-routing problem | Relaxed MIP |
| *Wang, Liu & Yang (2023)* | Genetic algorithm | Robust hub location and routing problem with a third-party logistics strategy | Random and greedy heuristics |
| *Wu, Qureshi & Yamada (2022)* | Adaptive large neighborhood search | Multiple (single) allocation hub location and routing problem | A two-stage greedy heuristic |
| *Bütün, Petrovic & Muyldermans (2021)* | Tabu search | Capacitated directed cycle hub location and routing problem under congestion | A three-stage decomposition-based greedy heuristic |
| *Yang, Bostel & Dejax (2019)* | Memetic algorithm | Hub location and routing problem with distinct collection and delivery tours | Random and greedy heuristics |
| *Danach, Gelareh & Neamatian Monemi (2019)* | Hyper heuristic | Capacitated single-allocation p-hub location and routing | Lagrangian relaxation technique |
| *Kartal, Krishnamoorthy & Ernst (2019)* | Iterated local search, Ant colony system, Discrete particle swarm optimization | Uncapacitated single allocation p-hub center and routing problem | Random |
| *Kartal, Hasgul & Ernst (2017)* | Ant colony system simulated annealing algorithm | Single allocation hub location and routing problem with simultaneous pick-up and delivery | For SA: A greedy heuristic drawn from the first solution of ant colony system |
| *Lopes et al. (2016)* | Biased random key genetic algorithm, Variable neighborhood search | A hub location and routing problem with one route per hub | Random and greedy heuristics |
| *Ratli et al. (2022)* | General variable neighborhood search | The same problem with *Lopes et al. (2016)* | Random |
| *Pandiri & Singh (2021)* | Hyper heuristic | The same problem with *Lopes et al. (2016)* | 2 Greedy Heuristics |
| *Rieck, Ehrenberg & Zimmermann (2014)* | Fix and optimize genetic algorithm | Many-to-many location- routing with inter-hub transport and multi-commodity pickup-and-delivery | Random initial solution with feasibility check |
| This study | Simulated annealing algorithm | Uncapacitated single allocation p-hub center and routing problem | -Random<br>-Greedy<br>-Hybrid<br>-Decomposition based mathematical programming formulation (Matheuristic) |

solution based on the Lagrangian relaxation technique. *Clarke & Wright (1964)* introduced three metaheuristic algorithms based on iterated local search, ant colony system, and discrete particle swarm optimization for the USApHCRP. The authors utilized random initial solutions for the iterated local search and discrete particle swarm optimization algorithm. *Lopes et al. (2016)* developed a biased random key genetic algorithm using both greedy-generated initial solutions and using random chromosomes for generating initial solutions for a hub location and routing problem. The authors also used a random solution for their variable neighborhood search-based heuristic. *Ratli et al. (2022)* addressed the

problem studied by *Lopes et al. (2016)* and proposed a solution in the form of the General Variable Neighborhood Search algorithm. The initial solution was randomly generated. *Pandiri & Singh (2021)* also considered the same problem studied by *Lopes et al. (2016)*. The authors developed two greedy heuristics for the initial solution to be used in the hyper heuristic. *Kartal, Hasgul & Ernst (2017)* utilized the initial solution obtained by the first iteration of an ant colony system algorithm in their study as the starting solution for the simulated annealing algorithm. Indeed, the final objective function results obtained by the simulated annealing algorithm are better than the final objective results obtained by the ant colony system. *Rieck, Ehrenberg & Zimmermann (2014)*, worked on many-to-may location routing problem with inter-hub transport and multi commodity pickup-and-delivery and for the solution of the problem, the authors developed a fix-and-optimize heuristic and a genetic algorithm.

As motivation drawn from real-life applications, readers can refer to the works of *Kuby & Gray (1993)*, *Aykin (1995)*, *Bruns, Klose & Stahly (2000)*, *Grünert & Sebastian (2000)*, *Wasner & Zapfel (2004)*, *Cetiner, Sepil & Sural (2010)*, *De Camargo, Miranda & Lokketangen (2013)*, and *Rodríguez-Martín, Salazar-González & Yaman (2014)* which incorporate routing decisions into the hub location problem for various versions of the problem. For interested readers, *Wandelt, Wang & Sun (2025)* provide a recent comprehensive review of hub location-routing problems (HLRPs), exploring the integration of hub location and vehicle routing to optimize logistics and transportation networks. The article examines various HLRP models, their mathematical formulations, and solution techniques, including exact and heuristic methods.

Recent algorithmic advancements in both metaheuristics and exact methods have led researchers to consider techniques from these two domains together. It has become common to integrate metaheuristic elements (typically local search) with exact algorithms devised for (mixed integer) linear programming. These techniques are called as matheuristics. *Archetti & Speranza (2014)* have categorized matheuristics into three distinct classes in routing related problems. The first of these is the decomposition approach, where the method subdivides the problem into smaller and less complex subproblems, each of which is addressed with a particular solution technique. The second class is improvement heuristics. In this type of matheuristics, the solution is improved by using mathematical programming models. The third class is branch-and-price/column generation-based approaches. These approaches are commonly used in solving routing problems. Such algorithms utilize a set partitioning formulation where a binary or integer variable is associated with each possible column. Given the exponential number of variables, the solution of the linear relaxation of the formulation is achieved through column generation. In the context of matheuristics based on branch-and-price or column generation, the exact method is adjusted to enhance convergence speed, albeit at the expense of losing optimality assurance. For example, the column generation phase may be terminated prematurely.

In the literature of the hub location and routing problem, there are few studies that include a matheuristic algorithm. *Aloullal, Saldanha-da-Gama & Todosijević (2023)* developed a four-phase matheuristic for the multi-period single-allocation hub

location-routing problem, combining principles of relax-and-fix. *Geçer (2020)* worked on fixed p-hub center and routing problem where the locations of the hubs are predetermined. The author suggested a mathematical programming formulation for the problem. For finding an initial solution with a mathematical model, we modified *Geçer*'s *(2020)* mathematical programming formulation to form the routes with a min-max multiple travelling salesman problem. Unlike *Geçer (2020)*, the locations of the hubs are not known *in priori*, and we find the hubs by using a modified radius-based p-hub center mathematical model and for creating the routes we use a modified mTSP formulation with min-max objective function to develop a matheuristic based strategy.

In this study, inspired by the concept of starting from a high-quality initial solution obtained through a constructive algorithm; an approach widely used in local search metaheuristics, we investigate the impact of various initial solution strategies on the final solution of the USApHCRP. To generate initial solutions, we propose four strategies: random, greedy, a hybrid approach combining random and greedy methods, and a decomposition-based matheuristic that formulates the problem in two hierarchical parts—hub location and node allocation, followed by routing part. These initial solutions serve as starting points for the proposed SA algorithm.

The second section presents the proposed initial solution strategies, detailing the random, greedy, hybrid, and decomposition-based matheuristic approaches, along with the modified formulations of the USApHCP and mTSP. The third section focuses on the specifics of the developed SA algorithm. The fourth section provides numerical results and their detailed analysis. The fifth section includes a discussion of the findings, and finally, the conclusion section summarizes the key insights and offers suggestions for future research.

## INITIAL SOLUTION STRATEGIES

USApHCRP involves determining hubs, assigning non-hub nodes to hubs, and creating the associated routing structure. The mathematical model for this problem was initially presented in the literature by *Geçer (2020)* in three formulations: two-index vehicle-flow, three-index vehicle-flow, and finally, the four-index multi-commodity flow-based formulation. For the mathematical model of the problem, references can be made to *Geçer (2020)* and *Kartal, Krishnamoorthy & Ernst (2019)*.

Before outlining the initial solution structures, a corresponding illustration of a hub location and routing example is presented in Fig. 1.

Figure 1 illustrates a hub location and routing structure. In this example, nodes 1 and 2 serve as hub nodes. Each route starts and ends at its respective hub node: the first hub follows the route 1 → 5 → 7 → 6 → 1, while the second hub follows the route 2 → 8 → 4 → 3 → 2. As observed, the routes involve multiple stopovers, with the primary objective being the minimization of the maximum distance or cost between any origin-destination pair. To accurately compute this maximum value, the model differentiates between collection and distribution routes for each vehicle. In many real-world scenarios, vehicles often return along the same path they took outbound, albeit in reverse order; thus, the outbound

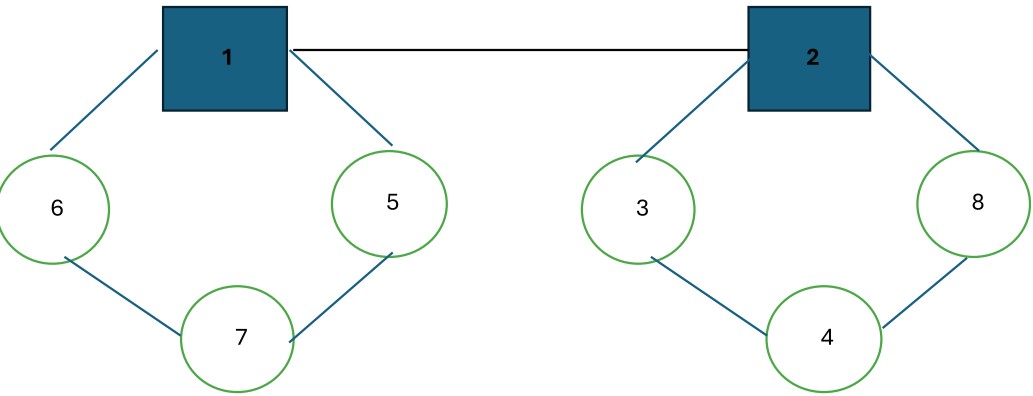

**Figure 1 Representation of a hub location and routing solution.**

journey is considered the collection route, while the return journey is treated as the distribution route.

### Random initial solution generation strategy

In the random initial solution strategy, hubs are chosen randomly and the assignments of non-hub nodes to the hubs are randomly performed to create routes.

### Greedy initial solution generation strategy

In this initial solution strategy, the location of the hubs and non-hub assignments to the hubs are carried out greedily. When selecting hubs, the distances from a node to all other nodes are calculated, and we select the required number of hubs by starting from the smallest. After selecting hub nodes, the allocation of nodes to the nearest hubs is performed for each vehicle, ensuring that each vehicle has a minimum of 1 and a maximum of $n - (p * (nv + 1)) + 1$ nodes. Finally, the routing process is accomplished by sequentially combining routes assigned to hubs.

### Random-greedy initial solution generation strategy

This initial solution strategy locates the hubs randomly, while the assignments of non-hub nodes to hubs are carried out greedily by allocating non-hub nodes to their nearest hubs. The routing process is completed similarly to the previous section by sequentially combining routes assigned to hubs.

### Mat-heuristic based initial solution generation strategy

This section presents a matheuristic algorithm developed to solve the single allocation p-hub center and routing problem, separately. Accordingly, the locations of hubs are determined by solving the modified radius based single allocation p-hub center problem. Then, depending on the number of vehicles assigned to each hub, either a modified min-max mTSP (if there are multiple vehicles assigned to each hub) or TSP formulations are utilized (if there is only one vehicle assigned per hub). In the subsequent section, modified single allocation p-hub center problem and min-max mTSP formulations are

introduced. We note here that although we use the following algorithm to construct an initial solution strategy, this approach could also be used as an entire solution finding strategy, since we could obtain a complete solution for the problem.

### Modified p-hub center problem formulation

In this study, to determine the hub locations, the radius-based p-hub center model proposed by *Ernst et al. (2009)* in the literature was used.

In the original version of the p-hub center problem, a node can be a hub, and no other non-hub nodes can be assigned to this hub, which means a cluster can be included in just one hub node. However, in our settings, at least a non-hub node has to be assigned to each vehicle. Therefore, the p-hub center problem's mathematical programming model developed by *Elatar, Abouelmehdi & Riffi (2023)* was modified by transforming it into a form where for each hub, at least the number of nodes assigned to that hub as vehicles will be assigned as nodes. Decision variables and parameters are used for this mathematical model as follows:

*Decision variables:*

$X_{ik} = \{1, \text{ if node } i \text{ is allocated to hub } k \ 0, \text{ otherwise}$
$X_{kk} = \{1, \text{ if node } k \text{ is a hub } 0, \text{ otherwise}$
$r_k = \text{radius of hub } k$

*Parameters:*

α: the discount coefficient,

$n_i$: the number of vehicles which are assigned to hub i

$d_{ij}$: distance between node $i \in N$ and node $j \in N$

We provide the modified mathematical modelling formulation:

$$Min \ Z, \tag{1}$$

$$Z \geq r_k + r_m + \alpha * d_{km}, \quad \forall \, i < m \in N, m \in N \tag{2}$$

$$\sum_{k \in N} x_{kk} = p, \tag{3}$$

$$\sum_{k \in N} x_{ik} = 1, \quad \forall \, i \in N \tag{4}$$

$$x_{ij} \leq x_{ii}, \forall \, i \in N, j \in N \tag{5}$$

$$r_k \geq d_{ik} * x_{ik}, \quad \forall \, i, k \in N \tag{6}$$

$$\sum_{i \in N} x_{ij} \geq (n_i + 1) * x_{jj}, \quad \forall \, j \in N \tag{7}$$

$$x_{ij} \in \{0, 1\}, \tag{8}$$

$$Z, r_k \in (0, 1). \tag{9}$$

The objective is represented by a free variable Z (Eq. (1)). The objective function aims to minimize the maximum distance between any pair of nodes, taking into account the associated hub radii. Constraint Eq. (2) ensures that the objective is no less than the travel distance between any pair of nodes allocated to hubs k and m. This is achieved by using rk and rm, which represent the distance to the farthest node allocated to hubs k and m, respectively. Constraint Eq. (3) ensures to locate of exactly p hubs, while Constraint Eq. (4) guarantees that each node is allocated to exactly one hub. Constraint Eq. (5) guarantees that if a node is allocated to another node, then the latter must be chosen as a hub. Constraint Eq. (6) calculates the radius of hub k. With Constraint Eq. (7), we ensure that a sufficient number of non-hub nodes are assigned to each hub.

As mentioned above, with this mathematical model, the non-hub nodes assigned to a hub and the hub locations are determined. Subsequently, a complete solution for the problem is obtained by transforming the m(TSP) model into a min-max objective function and solving it for observing the vehicle routes.

### Min-Max multiple travelling salesman problem formulation

In this section, we introduce a min-max mTSP. This mathematical model is developed by excluding the part of *Geçer*'s *(2020)* mathematical model that calculates the distance between hubs. By using the original version of *Geçer (2020)* mathematical model, the objective function can also be calculated by the model itself. However, it has been observed that this model can solve instances with up to 25 nodes in a reasonable time (*Geçer, 2020*). The solution of larger instances with more nodes cannot be solved in a reasonable time. Therefore, based on the decomposition approach, the mathematical model from *Geçer*'s *(2020)* study has been updated and used in this section for the case of a single hub with multiple vehicles assigned to each hub. This update involves minimizing the maximum distance between two nodes assigned to the same hub when multiple vehicles are assigned to the hub.

We note here that if there is only one vehicle assigned to each hub, then this problem can be solved *via* classical TSP mathematical model. Therefore, we used two-index vehicle flow TSP mathematical model if there is only one vehicle assigned to each hub.

The additional parameters and decision variables are as follows (Box 1):

| Box 1 | |
|---|---|
| *Sets* | *Indices* |
| *NH*: set of non-hub nodes | *k*: hub nodes |
| *H*: set of hub nodes | *i, j, l*: nodes that include hub nodes and non-hub nodes |
| *N*: set of nodes | |
| *V*: set of vehicles | *v,w*: vehicles |

*Parameters*

$h_i$: $\{1, node\ i\ is\ a\ hub\ node\ 0,\ otherwise.$

$L$: the maximum number of nodes a vehicle may visit.

*Decision Variables*

$cv_{kij}^{v} =$

$\{1,\ if\ vehicle\ v \in V\ begins\ its\ trip\ from\ hub\ node\ k \in H\ and\ travels\ to\ node\ i \in N\ and$ $node\ j \in N\ in\ that\ order\ (collection)\ 0,\ otherwise.$

$dw_{ijk}^{w} =$

$\{1, if\ vehicle\ w \in V\ completes\ its\ route\ at\ hub\ node\ k \in H\ by\ visiting\ node\ i \in N\ and$ $node\ j \in N\ in\ that\ order\ (distribution)\ 0,\ otherwise.$

$x_{ij}^{v} = \{1,\ if\ vehicle\ v \in V\ traverses\ arc\ (i,j)\ 0,\ otherwise$

$z_{iv} = \{1, if\ hub\ node\ i \in H\ is\ assigned\ to\ vehicle\ v \in V\ 0,\ otherwise.$

$mc_{kv}$ = maximum collection distance of vehicle $v \in V$ that starts its route from hub node $k \in H$.

$md_{kw}$ = maximum distribution distance of vehicle $w \in V$ that ends its route at hub node $k \in H$.

$u_i$ = auxiliary variable which defines the sequence number in which city $i \in N$ visited.

$\beta$ = the maximum distance between any node pair.

The mathematical programming formulation of our problem is shown below:

$$Min\ \beta \tag{10}$$

$$\sum_{i \in H} \sum_{j \neq i \in N} x_{ij}^{v} = 1 \qquad \forall v \in V, \tag{11}$$

$$\sum_{j \neq i} x_{ij}^{v} - \sum_{j \neq i} x_{ji}^{v} = 0 \qquad \forall i \in N, v \in V, \tag{12}$$

$$\sum_{j \neq i} \sum_{v} x_{ij}^{v} = n_i \qquad \forall i \in H, \tag{13}$$

$$\sum_{j \neq i} \sum_{v} x_{ji}^{v} = n_i \qquad \forall i \in H, \tag{14}$$

$$\sum_{j \neq i} \sum_{v} x_{ji}^{v} = 1 \qquad \forall i \in NH, \tag{15}$$

$$\sum_{j \neq i} \sum_{v} x_{ij}^{v} = 1 \qquad \forall i \in NH, \tag{16}$$

$$u_i - u_j + L * \sum_{v} x_{ij}^{v} + (L-2) * \sum_{v} x_{ji}^{v} \leq L - 1 \qquad \forall i \in NH, j \in NH: i \neq j, \tag{17}$$

$$\sum_{k \in H} cv_{kij}^{v} = x_{ij}^{v} \qquad \forall i \in N, j \in NH, v \in V: j \neq i, \tag{18}$$

$$cv_{kij}^{v} \leq x_{ij}^{v} \qquad \forall i \in N, j \in NH, k \in H, v \in V, \tag{19}$$

$$mc_{kv} = \sum_{i \in N} \sum_{j \in NH} (d_{ij} * cv_{kij}^{v}) \qquad \forall k \in H, v \in V, \tag{20}$$

$$\sum_{k\in H} dw_{ijk}^w = x_{ij}^w \qquad \forall\, i \in NH, j \in N, w \in V: j \neq i, \tag{21}$$

$$dw_{ijk}^w \leq x_{ij}^w \qquad \forall\, i \in NH, j \in N, k \in H, w \in V, \tag{22}$$

$$md_{iw} = \sum_{i\in NH} \sum_{j\in N} (d_{ij} * dw_{ijk}^w) \qquad \forall\, i \in H, w \in V, \tag{23}$$

$$\beta \geq (mc_{iv} + md_{iw}) \qquad \forall\, i \in H, v \in V, w \in V, \tag{24}$$

$$\beta \geq \sum_{i\in N} \sum_{j\neq i\in N} (d_{ij} * x_{ij}^v) - d_{kl} * x_{kl}^v - M * (1 - x_{kl}^v) \qquad \forall\, k \in N, l \in N, v \in V. \tag{25}$$

$$x_{ij}^v, dw_{ijk}^w, cv_{kij}^v \in \{0, 1\} \tag{26}$$

$$u_i, mc_{kv}, md_{iw}, \beta \geq 0 \tag{27}$$

The objective function minimizes the maximum distance between any node $i$ and node $j$ Eq. (10). Two different constraints form the objective function (Constraint Eq. (15) and Constraint Eq. (16)). The collection route is formed by picking up the parcels from all stopover nodes starting from the first non-hub node on the route through to the hub node, and the distribution route is formed by dropping off the parcels by visiting all non-hub nodes starting from the hub node through the last non-hub node along the route. Note here that unlike *Geçer (2020)*, we do not take into consideration the distances between hubs.

The main aim of the objective function is to minimize the longest distance between any pair of nodes, denoted as i and j Eq. (10). This objective function is derived from the combination of two constraints, namely Constraint Eq. (24) and Constraint Eq. (25). The creation of the collection route involves systematically picking up parcels from all intermediate nodes, starting at the initial non-hub node along the route and continuing up to the hub node. Conversely, the distribution route is formed by delivering parcels, which includes visiting all non-hub nodes. This process starts at the hub node and concludes at the final non-hub node along the vehicle route. Finally, the maximum of distance can be calculated by analyzing a single route where node i directly succeeds node j within the scope of a single hub (Constraint Eq. (25)).

Each vehicle is dedicated to a specific hub node (Constraint Eq. (11)). Constraint Eq. (12) enforces a degree restriction that guarantees an equivalent count of vehicles arriving and departing from each node i. Constraints Eqs. (13) and (14) guarantee that the count of outgoing links from a hub node i corresponds to the count of vehicles allocated to it. However, in the case where node i is a non-hub node, Constraints Eqs. (15), (16) ensure that a vehicle can make only one visit to each i − j non-hub link. Constraint Eq. (17) is the *Miller, Tucker & Zemlin (1960)*, (*Kartal & Ernst, 2015*) sub-tour breaking constraint that prevents sub-tours.

The collection distances (costs) for each vehicle route are calculated *via* Constraints Eqs. (18)–(20). Constraints Eqs. (18)–(20) calculate the collection distances for each vehicle. Constraint Eq. (18) ensures that the values of $cv_{kij}^v$ variables progress from the hub node k to the final non-hub node j along the route, incorporating any intermediate node i. The values of $cv_{kij}^v$ variables begin from the hub node k and extend to the final non-hub node j. We obtain the correct values $cv_{kij}^v$ variables for any solution of $x_{ij}^v$ variables if vehicle

*v* visits node i and node j in that order *via* Constraint Eq. (19). Through Constraint Eq. (20), the maximum collection distance using the $mc_{kv}$ variables along the vehicle route v for hub k is calculated. This distance represents the cumulative sum of all distances between the initial non-hub node and the hub node along the collection route of a given vehicle.

The distribution distances (costs) are calculated by Constraints Eqs. (21)–(23). Constraint Eq. (21) ensures that the $dw_{ijk}^w$ variables are assigned a value of 1 only when vehicle v completes its route at hub node k. Additionally, the $dw_{ijk}^w$ variables take on values starting from the initial non-hub node i when the final node k serves as a hub node. Constraint Eq. (22) ensures that the $dw_{ijk}^w$ variables attain accurate values based on the solutions of the $x_{ij}^w$ variables, when vehicle w sequentially visits node i and then node j. Constraint Eq. (23) ensures that the $dw_{ijk}^w$ variables attain accurate values based on the solutions of the $x_{ij}^w$ variables, when vehicle w sequentially visits node i and then node j. We determine $md_{kw}$ by aggregating the distances from the hub node to the most distant node along a vehicle's distribution route. Constraint Eq. (26) enforces the decision variables to take binary values. Lastly, Constraint Eq. (27) imposes non-negativity constraints on the variables.

## SIMULATED ANNEALING

Simulated annealing (SA) a stochastic search method originally proposed by *Kirkpatrick, Gelatt & Vecchi (1983)*, is employed to address optimization problems. It has been successfully applied to numerous large-scale real-world problems including TSP. Due to its ease of implementation, convergence characteristics, and the strategy it employs to avoid getting trapped in local optima. SA has become a widely used method for solving optimization problems in recent years. The name "Simulated Annealing" is derived from its analogy to the annealing process in solid materials. SA operates as a gradual refinement process. This progression involves the consideration not only of improved solutions but also the acceptance of poor solutions with a certain likelihood. This strategy leads the algorithm to escape from being trapped in local optima. This process constitutes one of the fundamental characteristics of the SA method. The acceptance probability is expressed as $e^{(-\Delta E/T)}$ based on a conceptual temperature. Here, $\Delta E$ represents the change in objective functions between the current solution and the generated neighboring solution. In this context, T represents the control parameter, which corresponds to the temperature. For small values of $\Delta E$, the likelihood of accepting a poor solution is higher than for larger values. Additionally, at higher temperature values, most new neighborhood solutions will be accepted. As the temperature approaches 0, the acceptance probability of new neighborhood solutions will decrease. Hence, in the SA algorithm, the initial temperature is typically set to a relatively high value to escape local optima.

| 1 | 6 | 7 | 5 | 1 | 2 | 3 | 4 | 8 | 2 |

**Figure 2  A solution structure for SA.**     

**Table 2  Example of each move applied to the initial state independently to obtain the perturbated solution/state.**

| No | State/Move type | Route 1 | Route 2 |
|----|-----------------|---------|---------|
| – | Initial state/solution | **1**-6-7-5-9-**1**; | **2**-3-4-8-**2**; |
| 1. | Inter-node-insertion | **1**-*7*-5-*6*-9-**1**; | **2**-3-4-8-**2**; |
| 2. | Intra-node-insertion | **1**-6-5-9-**1**; | **2**-3-4-*7*-8-**2**; |
| 3. | Inter-node-swap | **1**-*9*-7-5-*6*-**1**; | **2**-3-4-8-**2**; |
| 4. | Intra-node-swap | **1**-6-7-*8*-9-**1**; | **2**-3-4-*5*-**2**; |
| 5. | Inter-hub-swap | *7*-6-*1*-5-9-*7*; | **2**-3-4-8-**2**; |
| 6. | Inter-edge-opt | **1**-*5*-*9*-6-7-**1**; | **2**-3-4-8-**2**; |
| 7. | Intra-edge-opt | **1**-6-*4*-8-9-**1**; | **2**-3-*7*-*5*-**2**; |

**Note:**
The bold values in Table 2 indicate the hub locations. Italicized nodes indicate the ones that have undergone positional changes due to the corresponding neighborhood move operations.

## Simulated annealing neighborhood moves

The hub location and routing problem representation for the SA is structured as follows: In this solution representation, we selected the problem structure from Fig. 1 for clarity and ease of understanding for the reader. Accordingly, the bold numbers 1 and 2 represent the hubs, while the nodes adjacent to the hubs are the non-hub nodes included in the route, listed in the order they are visited is shown in Fig. 2.

As seen in Fig. 2, the routes start and end at hub nodes 1 and 2. In the route of hub 1, the nodes 6, 7, and 5 are visited sequentially, departing from the hub and returning to it (1 → 6 → 7 → 5 → 1). Similarly, in the route of hub 2, the nodes 3, 4, and 8 are visited in order, before returning to the hub node 2. In Table 2, we provide all SA neighborhood moves, demonstrating the independent application of each to the initial state to derive the perturbed solution/state for better clarity.

Table 2 summarizes the solution representation for the initial state (the first row), while the subsequent rows present the perturbed solution representations obtained after applying each move type independently to the initial state. The bold values in Table 2 indicate the hub locations. Italicized nodes indicate the ones that have undergone positional changes due to the corresponding neighborhood move operations. In the following, we explain how the neighborhood moves operate and provide examples for each.

1. Inter-node-insertion: A randomly selected node from a randomly selected route is placed next to another randomly chosen city within the same route. In routes containing only one node, both the node to be moved and the node where the operation will take place are the same, resulting in no change. Therefore, intra-insertion cannot be performed within routes containing only one non-hub node. An example of an insertion within the

same route is provided in the first row of Table 2, where node 6 has been inserted after node 5 within the same route (Route 1).

2. Intra-node-insertion: A randomly chosen node from one route is placed adjacent to a randomly selected node from another route (these two different routes may belong to the same hub), both of which are chosen randomly. If a non-hub node is chosen where there is only one non-hub node for relocation, the insertion is prohibited, it would result in only a hub node remaining in the route. An example of inter-route insertion is shown in the second row of Table 2, where node 7 has been relocated and positioned after the randomly chosen node 4 in a different vehicle's route.

3. Inter-node-swap: A route containing at least two cities is randomly selected, and two cities are then selected randomly from this route to swap their positions. The third row of Table 2. illustrates an example of a swap within the same route, where node 9 and node 6 have exchanged positions in Route 1.

4. Intra-node-swap: First, two routes are randomly selected. Then, one city is randomly chosen from each of these routes, and their positions are swapped. The fourth row of Table 2, presents an example of inter-route swap, where node 5 from the Route 1 and node 8 from the Route 2.

5. Inter-hub-swap: A node is randomly chosen from within a randomly selected route, and this node is located as the new hub, while the node that was previously the hub is placed at the location where the new hub is chosen. An example of the inter-hub swap is provided in fifth row of Table 2, where the hub node (node 1) has swapped positions with node 7 within its route.

6. Inter-edge-opt: Exchange the positions of two consecutive non-hub nodes within a single route. An example is provided in sixth row of Table 2, where two edges from hub 2's route have been swapped—specifically, edges 5–9 and 6–7 have been exchanged.

7. Intra-edge-opt: Swap the positions of two consecutive non-hub nodes (edges) between two routes that are randomly selected. An example of intra-edge-opt is provided in seventh row of Table 2, where the 4–8 edge from Route 2 has been swapped with the 7–5 edge from Route 1.

## The steps of SA

SA involves several parameters, including the initial temperature, cooling rate, acceptance criterion, and termination condition. The initial solution in this study is constructed by selecting one of the strategies presented in the previous section. The cooling process should be carried out slowly to obtain high-quality solutions, for which a cooling rate ($\theta$) is utilized. When transitioning from one temperature to another, the new temperature is determined by multiplying the previous temperature by the cooling rate ($T = \theta * T - 1$). Parameter selection is a crucial aspect for all heuristic optimization problems. A time limit is used as the stopping criterion for the algorithm. As the temperature value decreases, a certain number of new solutions are generated at each temperature to facilitate the search process.

We provide the steps for the SA that we used in this study:

Step 1: Input the parameters; T (initial temperature), θ (cooling rate), CS (number of solutions generated at each temperature), time_limit (maximum time limit).

Step 2: Generate an initial solution by choosing one of the four strategies (random, greedy, random-greedy and mathematical programming). Calculate f (cost of the initial solution).

Step 3: $S_{best}$ (best solution); S (current solution); S′ (neighbor solution); i_n (counter for solutions generated at each temperature) = 1, and $f_{best}$ (best solution cost); (current solution cost) = f; iter_n (iteration count); CS (iteration number at each temperature); time (elapsed time)=0;

Step 4: Generate a neighboring solution (S′) from the current solution as defined above. Calculate the cost of the neighboring solution (f′).

Step 5: Calculate Δ = f′ – f.

Step 6: If (Δ < 0), proceed to step 8; otherwise, go to step 7.

Step 7: Generate a uniformly distributed random number (r). If $r < e^{(-\Delta/T)}$, move to step 8; otherwise, proceed to step 9.

Step 8: Set S = S′ and f = f′.

Step 9: If f < $f_{best}$, then set $S_{best}$ = S and $f_{best}$ = f; otherwise, proceed to step 10.

Step 10: If i_n < CS, i_n = i_n + 1 and go back to step 4, otherwise proceed to step 11.

Step 11: Set iter_n = iter_n + 1, T + 1 = θ * T and i_n = 1.

Step 12: If time < time_limit, go to step 4; otherwise, proceed to step 13.

Step 13: Terminate the algorithm and write the results to a file.

## COMPUTATIONAL RESULTS

*Tan & Kara (2007)* introduced Turkish Cargo Delivery (Turkiye network) into the literature. There are 81 demand centers in this dataset, flow amongst these demand centers, and distances in this network. The Turkish network dataset is also available through the OR library (*Beasley, 1990*).

We employed SA with four different initial solution strategies independently to address the USApHCRP, and Gurobi Optimizer version 9.0.2 was utilized to run mathematical models. We conducted the experimental tests on a server operating with the UBUNTU operating system, equipped with a 3.5 GHz Intel Xeon E5-2643 v2 processor and a total of 128.863 GB RAM, with 30 GB allocated to the heap. The SA algorithm was implemented in Java, while the Gurobi components were implemented in Python. The code is available at https://github.com/AbdKa/pHMLRP.

### Parameters

The initial temperature T value which is set to 1,000,000 and the temperature is decreased by θ = 0.99; that is better than 0.9 and 0.95 after conducting a comparison on a decent sample. The maximum time for an instance is considered as the primary stopping condition for the algorithm. The run times for each strategy for 10 node instances are 10 s, for 15 nodes 30 s, for 25 nodes 60 s, for 50 nodes 270 s and for 81 nodes 1,000 s, respectively. The number of solutions that we generated at each iteration was n: number of

**Table 3 Results for Gurobi and SA with four initial solution strategies.**

| Gurobi | | | | | Simulated annealing | | | | | | | | |
| | | | | | RND | | GRD | | RND_GRD | | Matheuristic | |
| Problem | Obj | Lower bound | Gap (%) | CPU (s) | Gap (%) | CPU (s) | Gap (%) | CPU (s) | Gap (%) | CPU (s) | Gap (%) | CPU (s) |
|---|---|---|---|---|---|---|---|---|---|---|---|---|
| TR.10.2.1 | 3,597 | 3,597 | 0.00 | 124.05 | 0.00 | 10.0004 | 0.00 | 10.0023 | 0.00 | 10.0002 | 0.00 | 10.09 |
| TR.10.2.2 | 2,331 | 2,331 | 0.00 | 92.43 | 0.00 | 10.0001 | 0.00 | 10.0003 | 0.00 | 10.0002 | 0.00 | 10.08 |
| TR.10.3.1 | 2,651 | 2,651 | 0.00 | 75.73 | 0.00 | 10.0001 | 0.00 | 10.0003 | 0.00 | 10.0002 | 0.00 | 10.08 |
| TR.15.2.1 | 4,128 | 2,386 | 42.19 | 7.200 | 42.19 | 30.0001 | 42.19 | 30.0003 | 42.19 | 30.0011 | 42.19 | 30.23 |
| TR.15.2.2 | 2,835 | 1,652 | 41.72 | 7,200 | 39.4 | 30.0011 | 39.4 | 30.0029 | 39.4 | 30.0011 | 39.4 | 30.32 |

nodes (CS). We note here that the run time of each problem instance is independent of the initial solution time.

## Results

The numerical results were initially obtained using the GUROBI 9.0.2 solver. First, instances with 10 and 15 nodes were run. In Table 3, under the Gurobi column, the objective function, lower bound, gap between lower bound and objective function, and the CPU time spent for obtaining solutions by Gurobi are provided. It is important to mention that for the USApHCRP, we employed the two-index vehicle-flow formulation as outlined in the study by *Kartal, Krishnamoorthy & Ernst (2019)*. Under the column of the random initial solution strategy (RND), the gap to the lower bound recorded by Gurobi and the solution time which also includes the initial solution generation time, are presented. These values are subsequently given for the Greedy initial solution strategy (GRD), Random-Greedy (RND_GRD), and finally, for the mathematical programming based matheuristic algorithm.

We see in Table 3 that Gurobi was able to find optimal solutions for the datasets with 10 nodes. However, for the SA algorithm run with four different initial solution strategies, these values only slightly exceeded 10 s, and all solutions were able to reach optimal values. Notably, despite the matheuristic algorithm taking the longest time, the solution time for TR10.2.1 was 10.09 s. Even though 5 h were allocated for the 15.2.1 and 15.2.2 instances, they could not reach the optimal solution. For the 15.2.1 instance, both Gurobi and all initial strategies yielded the same result. Conversely, for the 15.2.2 instance, SA with four different initial strategies managed to achieve a slightly improved results compared to Gurobi. Moreover, due to Gurobi's inability to achieve optimal solutions within the allotted 5-h time frame, the decision was made not to run the other problem instances.

In the numerical results, the initial solution generation strategies were individually examined, and each problem instance was run 10 times. The initial solution strategy is the method used to find/construct a feasible whole solution. Tables 4–6 present the experimental results of/for different strategies in which an initial solution is constructed in a random, greed, hybrid of the previous two respectively. The initial solution strategy based on mathematical programming formulation (matheuristic) is provided in Table 7. In the first row of each table, the problem type is denoted as TR.n.p.v, where n signifies the

**Table 4  Results on random initial solution strategy with SA.** The footnotes indicate the optimal objective function values.

| Problem | Least cost | Avg. init. obj. gap | Avg. gap | Avg. iteration | Co of. variation | Init. sol. CPU (s) |
|---------|-----------|---------------------|----------|----------------|------------------|--------------------|
| TR.10.2.1 | 3,597* | 108.23 | 0.44 | 18,866 | 0.01 | 0.00036 |
| TR.10.2.2 | 2,331* | 163.23 | 0.00 | 19,039 | 0.00 | 0.00008 |
| TR.10.3.1 | 2,651* | 121.99 | 0.00 | 16,829 | 0.00 | 0.00006 |
| TR.15.2.1 | 4,128 | 169.28 | 1.59 | 49,867 | 0.02 | 0.00008 |
| TR.15.2.2 | 2,769 | 226.25 | 0.26 | 93,189 | 0.01 | 0.00109 |
| TR.25.2.1 | 5,247 | 269.05 | 1.28 | 173,390 | 0.02 | 0.00008 |
| TR.25.2.5 | 2,136 | 417.93 | 0.25 | 287,946 | 0.01 | 0.00017 |
| TR.25.5.1 | 2,710 | 374.58 | 2.62 | 198,719 | 0.02 | 0.00005 |
| TR.25.5.2 | 2,026 | 366.49 | 1.38 | 318,794 | 0.01 | 0.00007 |
| TR.50.2.1 | 7,861 | 400.39 | 2.82 | 974,202 | 0.02 | 0.00008 |
| TR.50.2.5 | 2,609 | 962.36 | 1.69 | 7,026,864 | 0.02 | 0.00015 |
| TR.50.5.1 | 3,737 | 681.86 | 1.48 | 1,581,300 | 0.02 | 0.00008 |
| TR.50.5.2 | 2,536 | 844.64 | 2.16 | 1,947,525 | 0.01 | 0.00011 |
| TR.81.2.1 | 9,980 | 507.36 | 2.63 | 2,822,916 | 0.02 | 0.00013 |
| TR.81.2.5 | 2,952 | 1,535.91 | 2.42 | 15,564,091 | 0.01 | 0.00028 |
| TR.81.5.1 | 4,514 | 975.43 | 2.55 | 6,725,910 | 0.02 | 0.00015 |
| TR.81.5.2 | 2,895 | 1,527.63 | 2.22 | 17,333,433 | 0.02 | 0.00027 |
| Average | 3,804.65 | 567.80 | 1.52 | 3,244,287 | 0.014 | 0.00019 |

**Table 5  Results on greedy initial solution strategy with SA.**

| Problem | Least cost | Avg. init. obj. gap | Avg. gap | Avg. iteration | Co of. variation | Init. sol. CPU (s) |
|---------|-----------|---------------------|----------|----------------|------------------|--------------------|
| TR.10.2.1 | 3,597 | 64.55 | 0.87 | 21,101 | 0.02 | 0.00230 |
| TR.10.2.2 | 2,331 | 87.69 | 0.00 | 20,310 | 0.00 | 0.00030 |
| TR.10.3.1 | 2,651 | 72.54 | 0.00 | 16,769 | 0.00 | 0.00030 |
| TR.15.2.1 | 4,128 | 87.79 | 1.04 | 43,107 | 0.02 | 0.00030 |
| TR.15.2.2 | 2,769 | 137.38 | 0.39 | 56,007 | 0.01 | 0.00290 |
| TR.25.2.1 | 5,247 | 170.67 | 2.41 | 145,965 | 0.02 | 0.00050 |
| TR.25.2.5 | 2,136 | 223.27 | 0.69 | 253,220 | 0.01 | 0.00060 |
| TR.25.5.1 | 2,721 | 226.39 | 1.77 | 200,990 | 0.01 | 0.00050 |
| TR.25.5.2 | 2,031 | 168.05 | 1.45 | 341,019 | 0.01 | 0.00070 |
| TR.50.2.1 | 7,797 | 256.59 | 2.92 | 1,411,573 | 0.02 | 0.00080 |
| TR.50.2.5 | 2,597 | 643.09 | 1.88 | 6,760,667 | 0.01 | 0.00110 |
| TR.50.5.1 | 3,709 | 456.51 | 3.78 | 1,599,012 | 0.03 | 0.00070 |
| TR.50.5.2 | 2,569 | 576.96 | 1.42 | 3,873,652 | 0.02 | 0.00090 |
| TR.81.2.1 | 9,823 | 310.99 | 4.29 | 2,632,723 | 0.02 | 0.00100 |
| TR.81.2.5 | 2,906 | 1,139.81 | 2.80 | 17,689,600 | 0.02 | 0.00160 |
| TR.81.5.1 | 4,530 | 702.49 | 2.76 | 6,009,446 | 0.02 | 0.00250 |
| TR.81.5.2 | 2,896 | 1,091.02 | 3.23 | 11,885,840 | 0.02 | 0.00260 |
| Average | 3,790.47 | 377.40 | 1.86 | 3,115,353 | 0.0153 | 0.00115 |

**Table 6 Results on random-greedy initial solution strategy with SA.**

| Problem | Least cost | Avg. init. obj. gap | Avg. gap | Avg. iteration | Co of. variation | Init. sol. CPU (s) |
|---|---|---|---|---|---|---|
| TR.10.2.1 | 3,597 | 57.24 | 0.44 | 19,626 | 0.01 | 0.0002 |
| TR.10.2.2 | 2,331 | 88.59 | 0.00 | 19,286 | 0.00 | 0.0002 |
| TR.10.3.1 | 2,651 | 79.25 | 0.00 | 19,889 | 0.00 | 0.0002 |
| TR.15.2.1 | 4,128 | 87.79 | 2.99 | 44,082 | 0.02 | 0.0011 |
| TR.15.2.2 | 2,769 | 141.96 | 0.92 | 57,351 | 0.02 | 0.0011 |
| TR.25.2.1 | 5,247 | 126.22 | 0.86 | 162,182 | 0.01 | 0.0002 |
| TR.25.2.5 | 2,136 | 260.53 | 1.15 | 384,528 | 0.01 | 0.0004 |
| TR.25.5.1 | 2,710 | 199.96 | 2.00 | 233,931 | 0.01 | 0.0004 |
| TR.25.5.2 | 2,042 | 171.11 | 1.12 | 313,575 | 0.01 | 0.0005 |
| TR.50.2.1 | 7,919 | 175.81 | 2.37 | 701,317 | 0.02 | 0.0003 |
| TR.50.2.5 | 2,571 | 483.08 | 2.22 | 5,385,418 | 0.01 | 0.0015 |
| TR.50.5.1 | 3,696 | 326.14 | 2.58 | 1,498,131 | 0.02 | 0.0007 |
| TR.50.5.2 | 2,547 | 445.19 | 1.83 | 3,307,866 | 0.02 | 0.0008 |
| TR.81.2.1 | 9,922 | 263.16 | 3.68 | 3,759,052 | 0.02 | 0.0003 |
| TR.81.2.5 | 2,946 | 909.78 | 1.76 | 37,548,078 | 0.01 | 0.0011 |
| TR.81.5.1 | 4,495 | 558.62 | 2.71 | 8,360,778 | 0.02 | 0.0011 |
| TR.81.5.2 | 2,913 | 756.40 | 2.14 | 20,706,028 | 0.01 | 0.0015 |
| Average | 3,801.18 | 301.81 | 1.69 | 4,854,183 | 0.0129 | 0.0007 |

**Table 7 Results on matheuristic algorithm.**

| Problem | Least cost | Avg. init. obj. gap | Avg. gap | Avg. iteration | Co of. variation | Init. sol. CPU (s) |
|---|---|---|---|---|---|---|
| TR.10.2.1 | 3,597 | 47.71 | 0.00 | 20,786 | 0.00 | 0.09 |
| TR.10.2.2 | 2,331 | 71.13 | 0.00 | 20,310 | 0.00 | 0.08 |
| TR.10.3.1 | 2,651 | 51.83 | 0.00 | 16,769 | 0.00 | 0.08 |
| TR.15.2.1 | 4,128 | 34.35 | 0.95 | 46,585 | 0.01 | 0.23 |
| TR.15.2.2 | 2,769 | 50.23 | 0.00 | 51,277 | 0.00 | 0.32 |
| TR.25.2.1 | 5,266 | 28.96 | 2.99 | 136,261 | 0.02 | 0.75 |
| TR.25.2.5 | 2,136 | 23.36 | 0.60 | 333,694 | 0.01 | 45,268 |
| TR.25.5.1 | 2,721 | 31.20 | 1.86 | 217,771 | 0.01 | 25,934 |
| TR.25.5.2 | 2,031 | 54.26 | 0.66 | 356,891 | 0.01 | 45,263 |
| TR.50.2.1 | 7,912 | 16.29 | 2.57 | 2,762,613 | 0.02 | 32,021 |
| TR.50.2.5 | 2,572 | 30.87 | 3.24 | 2,992,098 | 0.02 | 14,411.21 |
| TR.50.5.1 | 3,696 | 37.61 | 3.53 | 2,027,856 | 0.03 | 51.11 |
| TR.50.5.2 | 2,521 | 64.22 | 2.53 | 2,319,834 | 0.02 | 176.11 |
| TR.81.2.1 | 9,846 | 14.91 | 5.60 | 3,774,488 | 0.03 | 40.92 |
| TR.81.2.5 | 2,905 | 33.29 | 3.55 | 21,561,828 | 0.02 | 14,436.73 |
| TR.81.5.1 | 4,502 | 39.94 | 2.42 | 12,002,209 | 0.01 | 5,235.67 |
| TR.81.5.2 | 2,853 | 103.33 | 4.42 | 19,103,802 | 0.02 | 3,127.45 |
| Average | 3,790.41 | 43.15 | 2.05 | 3,984,887 | 0.0135 | 10,939.22 |

number of nodes, p represents the number of hubs, and v indicates the number of vehicles allocated to each hub. The second column of each table presents the least cost objective function value obtained by the respective initial solution strategy. The third column records the average gap between the 10 generated initial solutions and the best solution yielded by that particular initial solution strategy. The fourth column of the tables gives the average gap; average iteration numbers, coefficient of variance and lastly, the average time spent for generation the initial solutions in fifth, sixth and seventh columns, respectively. The footnotes in Table 4 indicate the optimal objective function values. They are not marked in the other tables to keep the representation simple.

When we examine Table 4, the average of the best solutions reached by the algorithm, wherein the initial solutions are generated randomly, is 3,804.65; however, this value is 4,036.35 in *Kartal, Krishnamoorthy & Ernst (2019)* study. Indeed, we believe that this disparity highlights the efficacy of the SA algorithm. In Table 4, we observe that the algorithm consistently attains the optimal objective function values across all 10 runs for the TR10.2.2 and TR10.3.1 instances. The largest average gap value of the SA algorithm, initiated with random solution strategy, remains at a deviation of 2.82% from the best objective function value (TR50.2.1). Subsequently, the second maximum average gap value emerges in TR81.2.1 (2.63%). As can be seen in Table 4, the average iteration counts increase proportionally to the complexity of the problem. Among the randomly generated initial solutions, the most substantial differences are observed in the TR81.2.5 and TR81.5.2 instances, respectively. This can be attributed to the observation that as the number of nodes and routes increases, random solutions tend to deviate significantly from the optimal solution values. However, for the problem instance TR10.2.1, featuring 10 nodes and two routes, the initial solution's gap to the optimal solution is the closest on average, at 108.23%. The overall average gap of the initial objective function values to the best solutions is around 567%. This represents the highest overall gap value among the provided initial solutions in this research.

We can observe from Table 5 that the average of the best results achieved by the greedy initial solution strategy with SA is 3,790.47. However, SA with random initial solution strategy corresponds to 3,804.65, as indicated in Table 4. Analogous to the randomly constructed initial solution strategy, in the instances where the initial solution is formed greedily, the largest gap is 1,139.81% in TR81.2.5 instance in terms of first solution. We think that this huge gap stems from locating of hubs in distant nodes from the optimal hub locations. Analyzing final solutions', the average gaps, we obtain that the highest gap is found in the TR81.2.1 instance (4.29%). Here, we believe that where we locate the hubs significantly influences the solution due to the there is only single route in per hub. Nevertheless, overall average gap remains at 1.86%. Considering the robustness of the algorithm, we anticipate that this value is quite reasonable. We note here that since there is no randomness in generating the initial solutions by using greedy strategy, we report the initial objective functions gaps considering the best-found solutions by this strategy (Table 5). Lastly, when we examine the CPU times of the initial solutions, we observe that the runtime of this deterministic algorithm is quite short.

When examining the hybrid constructed random-greedy initial solution strategy (Table 6), we obtain that the average of the best solution values (3,801.18) falls within the range between the averages of solutions initiated with random solutions (3,804.65) and those with greedy solutions (3,790.47), respectively. We see that this strategy outperforms the previous two initial solution strategies (random and greedy), with overall average gap of approximately 301.81% in terms of the first (initial) solutions. Upon investigating the average gaps, in line with the previous tables (Tables 4 and 5), the highest gap is observed in TR81.2.1 (3.68%). Furthermore, the algorithm's average run time of generating initial solutions takes approximately 0.0007 s.

In Table 7, we present the results of the matheuristic algorithm, which employs a strategy relying solely on mathematical models as the initial solution generating approach, constituting the method undertaken in this part. It's noteworthy that, the data in the second column is deterministic. TR10.2.1 and TR15.2.2 consistently found the optimal solution in all 10 runs (Table 7). This might be because the success of the hub changing operator; it could find the optimal locations of the hubs in each run. We note here that if the algorithm cannot place the hubs into the exact nodes, then achieving the optimal solution is impossible. When we analyzed the overall average gap of this strategy (2.05%), a slight increase is noticeable in comparison to other strategies. The reason for this situation can be explained that the random neighborhood changes of the heuristic algorithm, which starts from a very good place as a starting point, may not cause quite large jumps on the solution to get rid of stucking in local optima.

The process of generating mathematical programming based initial solutions requires a substantial amount of computational time, particularly if the problem instances have a large number of nodes $n$, (denoted as $n$). This is due to the time constraints we put on the p-hub center model (2 h in total), and an additional 2 h dedicated to finding each route for each hub through mTSP or TSP formulations. To illustrate, instances like TR.50.2.5 and TR.81.2.5 each extend beyond the 4-h threshold (two vehicles * 2 h). Consequently, the optimization of the two hubs' routes utilizing mTSP, where the number of vehicles per hub ($v$) exceeds 1, hits the predetermined time limit in both scenarios, with the remaining CPU time allocated to p-hub center optimization. However, when considering the practical significance of the problem in real-world scenarios, it is believed that the spent time for the initial solution generation and the algorithm are tolerable, especially when taking into account the cost reduction achieved.

We also analyze the coefficient of variation values as presented in Tables 4 to 7. The coefficient of variation (CoV) is a statistical measure used to evaluate the relative dispersion of a dataset by expressing the standard deviation as a proportion of the mean, simply calculated by dividing the standard deviation by the mean. It provides insights into the stability and consistency of an algorithm's performance across multiple runs. In this study, CoV values have been analyzed for four different initialization strategies within SA. Table 4, which reports the random initial solution strategy, presents the lowest CoV values, ranging from 0.00 to 0.02, with an average of 0.014. This suggests that although random initialization introduces some level of uncertainty, the SA method effectively stabilizes the final solutions, leading to minimal variability across different runs. Table 5, in which we

Table 8 **A comparison of results.** All values shown in bold under the "Least cost" columns represent the minimum objective function values obtained up to that point for the corresponding problem instance. Please note that the values reported for TR.10.2.1, TR.10.2.2, and TR.10.3.1. represent optimal solutions.

| Problem | Kartal, Krishnamoorthy & Ernst (2019) Least cost | Imp. (%) | Least cost | Initial solution strategies | | | | | | | | | | | |
| --- | --- | --- | --- | --- | --- | --- | --- | --- | --- | --- | --- | --- | --- | --- | --- |
| | | | | RND | | | GRD | | | RND-GRD | | | Matheuristic | | |
| | | | | Avg. init. obj. F. | Least cost | Min gap (%) | Avg. init. obj. F. | Least cost | Min gap (%) | Avg. init. obj. F. | Least cost | Min gap (%) | Init. obj. F. | Least cost | Min gap (%) |
| TR.10.2.1 | **3,597** | 0.0000 | **3,597** | 7,490 | **3,597** | 0.0000 | 5,919 | **3,597** | 0.0000 | 5,656 | **3,597** | 0.0000 | 5,310 | **3,597** | 0.0000 |
| TR.10.2.2 | **2,331** | 0.0000 | **2,331** | 6,136 | **2,331** | 0.0000 | 4,375 | **2,331** | 0.0000 | 4,396 | **2,331** | 0.0000 | 3,989 | **2,331** | 0.0000 |
| TR.10.3.1 | **2,651** | 0.0000 | **2,651** | 5,885 | **2,651** | 0.0000 | 4,574 | **2,651** | 0.0000 | 4,752 | **2,651** | 0.0000 | 4,025 | **2,651** | 0.0000 |
| TR.15.2.1 | **4,128** | 0.0000 | **4,128** | 11,116 | **4,128** | 0.0000 | 7,752 | **4,128** | 0.0000 | 7,752 | **4,128** | 0.0000 | 5,546 | **4,128** | 0.0000 |
| TR.15.2.2 | **2,769** | 0.0000 | **2,769** | 9,034 | **2,769** | 0.0000 | 6,573 | **2,769** | 0.0000 | 6,700 | **2,769** | 0.0000 | 4,160 | **2,769** | 0.0000 |
| TR.25.2.1 | **5,247** | 0.0000 | **5,247** | 19,364 | **5,247** | 0.0000 | 14,202 | **5,247** | 0.0000 | 11,870 | **5,247** | 0.0000 | 6,791 | 5,266 | 0.0036 |
| TR.25.2.5 | 2,219 | 0.0389 | **2,136** | 11,063 | **2,136** | 0.0000 | 6,905 | **2,136** | 0.0000 | 7,701 | **2,136** | 0.0000 | 2,635 | **2,136** | 0.0000 |
| TR.25.5.1 | 2,779 | 0.0255 | **2,710** | 12,861 | **2,710** | 0.0000 | 8,881 | 2,721 | 0.0041 | 8,129 | **2,710** | 0.0000 | 3,570 | 2,721 | 0.0041 |
| TR.25.5.2 | 2,112 | 0.0424 | **2,026** | 9,451 | **2,026** | 0.0000 | 5,444 | 2,031 | 0.0025 | 5,536 | 2,042 | 0.0079 | 3,133 | 2,031 | 0.0025 |
| TR.50.2.1 | 7,919 | 0.0156 | **7,797** | 39,336 | 7,861 | 0.0082 | 27,803 | **7,797** | 0.0000 | 21,841 | 7,919 | 0.0156 | 9,201 | 7,912 | 0.0147 |
| TR.50.2.5 | 3,003 | 0.1680 | **2,571** | 27,717 | 2,609 | 0.0148 | 19,298 | 2,597 | 0.0101 | 14,991 | **2,571** | 0.0000 | 3,366 | 2,572 | 0.0004 |
| TR.50.5.1 | 4,233 | 0.1453 | **3,696** | 29,218 | 3,737 | 0.0111 | 20,641 | 3,709 | 0.0035 | 15,750 | **3,696** | 0.0000 | 5,086 | **3,696** | 0.0000 |
| TR.50.5.2 | 2,970 | 0.1781 | **2,521** | 23,956 | 2,536 | 0.0060 | 17,391 | 2,569 | 0.0190 | 13,886 | 2,547 | 0.0103 | 4,140 | **2,521** | 0.0000 |
| TR.81.2.1 | 10,002 | 0.0182 | **9,823** | 60,615 | 9,980 | 0.0160 | 40,372 | **9,823** | 0.0000 | 36,033 | 9,922 | 0.0101 | 11,314 | 9,846 | 0.0023 |
| TR.81.2.5 | 3,515 | 0.2100 | **2,905** | 48,292 | 2,952 | 0.0162 | 36,029 | 2,906 | 0.0003 | 29,748 | 2,946 | 0.0141 | 3,872 | **2,905** | 0.0000 |
| TR.81.5.1 | 5,568 | 0.2387 | **4,495** | 48,545 | 4,514 | 0.0042 | 36,353 | 4,530 | 0.0078 | 29,605 | **4,495** | 0.0000 | 6,300 | 4,502 | 0.0016 |
| TR.81.5.2 | 3,575 | 0.2531 | **2,853** | 47,120 | 2,895 | 0.0147 | 34,492 | 2,896 | 0.0151 | 24,947 | 2,913 | 0.0210 | 5,801 | **2,853** | 0.0000 |
| | **Average** | **0.0785** | | | | 0.0054 | | | 0.0037 | | | 0.0047 | | | 0.0017 |

provide the results for the greedy initial solution strategy, shows a slight increase in variability, with CoV values ranging between 0.00 and 0.03 and an average of 0.0153. While greedy approach results in obtaining better initial solutions, it also leads to slightly higher variation in the final outcomes, possibly due to premature convergence to locally optimal solutions. The random-greedy hybrid approach, analyzed in Table 6, aims to strike a balance between randomness and greediness. The results indicate that this method achieves an average CoV of 0.0129, which is lower than the random or greedy strategies. This suggests that incorporating elements of both randomness and greediness can improve solution stability while still benefiting from the strengths of each approach. Finally, in Table 7, we provide the results for the matheuristic algorithm, exhibits CoV values in a similar range (0.00 to 0.03) but shows slightly higher variation in larger problem instances. The average CoV of 0.0135 suggests that while the matheuristic approach maintains strong consistency, starting from good initial solutions can sometimes cause the algorithm to get trapped in local optima, leading to slightly increased variability in some cases.

Overall, the results confirm that all four initialization strategies exhibit low CoV values, demonstrating the stability and reliability of the SA algorithm. While random initialization

provides general robustness, greedy and hybrid strategies lead to slightly improved initial solutions but with minor increases in variability. The matheuristic approach, despite being more computationally demanding, maintains comparable consistency levels. These findings suggest that the choice of initialization strategy should be guided by the problem characteristics, with hybrid and matheuristic approaches being particularly advantageous for larger and more complex instances due to their balance between solution quality and stability.

Lastly, we present Table 8 to comprehensively evaluate the overall effectiveness of all initial solution strategies. Note here that the time limits employed in *Kartal, Krishnamoorthy & Ernst (2019)* were also adopted in this study. We acknowledge that to perform a fair comparison, obtaining all runs under the same environment is necessary. Currently, our settings are not identical; however, considering the significance of the problem, certain variations in the settings can be tolerated.

In Table 8, the problem instance is presented first, followed by the best results obtained from *Kartal, Krishnamoorthy & Ernst (2019)* study. In columns 3 and 4, the improvement rates and best objective function values obtained by the SA (within all strategies) are presented, respectively. Below the initial solution strategies, each column provides the average initial objective function values, the least objective function values found by each strategy, and finally, the minimum gap of each strategy's results to the best values found by the SA algorithm. In the Table 8, all values shown in bold under the "Least cost" columns represent the minimum objective function values obtained up to that point for the corresponding problem instance. Please note that the values reported for TR.10.2.1, TR.10.2.2, and TR.10.3.1. represent optimal solutions.

In Table 8, 11 out of 17 instances, ranging from TR25.5.2 to TR81.5.2, we observed better (less) objective function values. When compared to the study by *Kartal, Krishnamoorthy & Ernst (2019)*, the overall average improvement rate is 7.85%. The minimum improvement rate is observed in the TR50.2.1 instance (1.56%), while the highest improvement rate is recorded in the TR81.5.2 instance (25.31%). The maximum improvement rate is recorded in the TR81.5.2 instance (25.31%). Remarkably, instances denoted by TRxx.2.1 show the least pronounced improvement rates. It is noteworthy that a slight improvement rate is also noted in the TR81.2.1 instance (1.82%). The relatively lower improvement rates in instances with two nodes can be explained by the substantial route costs. Since the route costs are higher, even though a notable improvement is achieved, the increase remains relatively low.

When we analyzed Table 8 further, it is observed that as the number of nodes and vehicles increases, the improvement percentages also rise. For instance, the problem instance where the first improvement is observed in TR25.2.5, the improvement rate stands at 3.89%, whereas in the TR81.5.2 instance, this improvement percentage is approximately six times higher. Furthermore, among the instances with 81 nodes, improvement percentages were above 20% in three cases, except TR81.2.1 instance.

Interestingly, it is observed in Table 8 that an improvement pattern is observed across all initial solution construction strategies. SA algorithm starting with a random initial solution has managed to find better solutions in the problems with lower node numbers

(25 nodes); namely TR25.5.1, TR25.2.5, and TR25.5.2. On the other hand, SA starting with a greedily constructed initial solution strategy has achieved the most significant improvements in instances denoted by TRxx.2.1; such as TR50.2.1 and TR81.2.1. Random-Greedy initial solution strategy demonstrated the most substantial improvements in the instances denoted by TRxx.5.1 and TRxx.2.5. Lastly, the matheuristic produced the most significant improvement percentages across instances containing 50 and 81 nodes, which are relatively considered large nodes.

The strategy that yielded the maximum of overall minimum average gap was SA started with a random solution strategy (0.54%). This result suggests that a guided initial solution strategy might result in further performance increase in the algorithm. Random-Greedy initial solution strategy ranks second in terms of improvement percentage, with an overall minimum average gap value of approximately 0.47%. This observation is in line with our expectations. Greedy initial construction solution strategy was in third order in terms of the improvement rate (0.37%), while the closest algorithm in terms of the overall minimum gap to the best solutions found was the matheuristic algorithm (0.17%). In conclusion, in larger-node instances, the most significant improvements are achieved by the matheuristic approach. Despite the time-consuming nature of this initial solution generation strategy, the matheuristics' success increases as the node count increase, making it a viable choice for decision-makers in the logistics sector.

To evaluate the performance of SA, in this study, we employed the lower bound used in the *Kartal, Krishnamoorthy & Ernst (2019)* study. The lower bound involves using the maximum distance of the associated data instance. Distances between node i and node j, with at least one intermediary node k, where $i \neq j \neq k$, are computed utilizing the formula $d_{ik} + d_{kj}$. According to the principle of the triangle inequality, it is established that $d_{ik} + d_{kj} \geq d_{ij}$ for all i, j, and k. Given that each $d_{ij}$ in the distance dataset includes the maximum value among $d_{ij}$, this particular value serves as a lower limit for USApHCRP. The lower bound values are 1,740 for instances with 25 nodes, and 2,045 for instances with 50 and 81 nodes, respectively. For SA, the average gaps between its best solution costs and the corresponding lower bounds were found to be 33.45% for instances with 25 nodes, 38.75% for instances with 50 nodes, and 47.58% for instances with 81 nodes, respectively.

## DISCUSSION

In this section, we discuss the advantages and disadvantages of our proposed methodology, highlighting key trade-offs. This discussion not only revisits our primary research objectives but also contextualizes our findings within the broader literature.

Our study has several key strengths. First, we show that tailored initial solutions can greatly improve solution quality within a single-trajectory metaheuristic, specifically the SA algorithm.

Our initial solution strategies integrate problem-specific techniques from matheuristics and greedy heuristics, drawing on expertise knowledge. This approach takes advantage of the problem's decomposable structure, resulting in good solutions.

Furthermore, our methods exhibit strong performance on large-scale problem instances. While NP-hard problems are inherently challenging, employing matheuristic-based and greedy initial solution strategies provide a competitive advantage over established methods, as demonstrated in our comparison with *Kartal, Krishnamoorthy & Ernst (2019)*.

Notwithstanding the advantages, the matheuristic initial solution approach also has some limitations. The main drawback is the significant computational time required to solve mixed-integer programming models using a commercial solver to achieve optimal solutions, especially for large-scale problems. As the problem size increases, the computational time required on the solver grows, prompting us to impose a 2-h time limit on the solution process. While this constraint ensures practical computation times, it creates a trade-off between solution quality and efficiency.

Additionally, since SA is a single-trajectory-based metaheuristic that perturbs solutions through neighborhood moves and often accepts worse solutions, especially in the early stages, the impact of the initial solution may diminish as the algorithm progresses. This effect is particularly evident in the case of random initial solutions, where despite starting from poor-quality solutions, the final objective function values remain competitive. However, this may not completely eliminate the benefits of starting from a good initial solution, such as those generated by matheuristic or greedy approaches. On the contrary, starting from a well-structured solution is likely to have a positive impact, particularly in increasing the chances of outperforming existing methods in the literature.

## CONCLUSIONS

This study addresses the uncapacitated single allocation p-hub center and routing problem by proposing an SA algorithm enhanced with four distinct initial solution strategies. Our findings suggest that the quality and structure of the initial solution can significantly impact the performance of an improvement-type metaheuristic, particularly across different problem structures, such as varying hub numbers and vehicle configurations. Our analysis indicates that random initialization is more effective for smaller problem instances, while greedy-based initialization performs better in cases with fewer hubs. This might be because the greedy approach selects hubs based on proximity to other nodes, often selecting more central locations. When the number of hubs is low, this approach might lead to a more balanced distribution of routes. The random-greedy strategy effectively balances exploration and exploitation, producing strong results across multiple instances. In contrast, the matheuristic-based strategy yields the most significant gains in larger problem instances. Specifically, the matheuristic approach employs the decomposition of the problem, allowing the underlying mathematical models to find either exact solutions or high-quality approximations within the given time constraints.

Our research contributes to the literature in two folds. First, it systematically examines how different initial solution generation strategies affect the performance of a single-trajectory metaheuristic, filling an existing gap where most studies have focused on population-based methods. Second, it introduces a novel matheuristic algorithm for the

USApHCRP, which utilizes a modified p-hub center and an mTSP-based routing formulation, demonstrating the potential of hybrid methodologies in complex network optimization problems.

The initial solution strategies developed in this study are problem-specific, constructive-based algorithms that incorporate expert knowledge. This underscores the importance of domain expertise in designing effective constructive heuristics for decomposed problem structures. The integration of expert-driven insights into constructive solution generation could significantly influence the final outcomes of an optimization algorithm. Given its potential advantages, the methodologies developed in this study may also be applicable to similar problem structures where location-routing decisions play a crucial role.

The computational results on the Turkish Network validate the effectiveness of our approach, as our SA algorithm outperformed other metaheuristics in the literature. These promising results encourage further exploration into the use of constructive heuristics within metaheuristic frameworks.

Future work could extend our methodology to other variants of the hub location and routing problem, such as capacitated or multiple allocation versions, and explore additional hybrid strategies that integrate metaheuristic and exact optimization techniques. Expanding these results to other domains, it is evident that any problem with an underlying location-routing model could benefit from this approach. Additionally, future research may compare the performance of single-trajectory and population-based algorithms for this decomposable network design problem, broadening the applicability of our findings and contributing to the ongoing development of efficient solution methods for combinatorial optimization challenges.

## ACKNOWLEDGEMENTS

We used ChatGPT to assist with the translation of a small portion of this manuscript from Turkish to English and for minor linguistic refinement. However, all intellectual contributions, interpretations, and conclusions remain the sole responsibility of the authors.

### Funding

The authors received no funding for this work.

### Competing Interests

The authors declare that they have no conflict of interest.

### Author Contributions

- Abdul Kader Kassoumeh conceived and designed the experiments, performed the experiments, analyzed the data, performed the computation work, prepared figures and/or tables, authored or reviewed drafts of the article, and approved the final draft.

- Zühal Kartal conceived and designed the experiments, performed the experiments, analyzed the data, prepared figures and/or tables, authored or reviewed drafts of the article, and approved the final draft.
- Ahmet Arslan conceived and designed the experiments, performed the experiments, analyzed the data, prepared figures and/or tables, authored or reviewed drafts of the article, and approved the final draft.

## Data Availability

The code and data is available at GitHub and Zenodo:

- https://github.com/AbdKa/pHMLRP.

- Abdul Kader Kassoumeh, Ahmet Arslan, & zuhalk. (2025). AbdKa/pHMLRP: new release (v1.0.0). Zenodo. https://doi.org/10.5281/zenodo.15489335.

The data is also available at GitHub: https://github.com/AbdKa/pHMLRP/tree/master/TR.

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
