# Peer review of "The effect of different initial solutions on the metaheuristic algorithms for the single allocation p-hub center and routing problem"

_PeerJ Computer Science, doi:10.7717/peerj-cs.2840_

## Round 0.1 · original submission · Major Revisions

Dear author,

We have received two reports for your paper, and you will see from the comments that the reviewers have proposed detailed comments on your paper. The main questions that you should respond are as follows:

- Explain the reason for choosing the SA algorithm and explain its advantages and limitations in comparison with the existing metaheuristic algorithms;

- The second point is to present a deep literature review and to define the gap. You will see that the reviewer has proposed several references to read it. I have checked these references, and it looks like you could use them for discussion regarding the previous concerns;

- The final question is the validation of the results. I am expecting to see more detailed validation with statistical analysis.

Along with the revised paper, I am expecting to see the point-to-point letter with a response to all comments.

Regards,
Editor

**Language Note:** The review process has identified that the English language must be improved. PeerJ can provide language editing services - please contact us at [email protected] for pricing (be sure to provide your manuscript number and title). Alternatively, you should make your own arrangements to improve the language quality and provide details in your response letter. – PeerJ Staff

Reviewer 1 ·

Basic reporting

The submitted paper is generally well written and has contribution in the combinatorial optimization field. However, hoping to assist the authors in their research efforts, I provide several suggestions for improving the presented work:
- Abstract - The abstract needs to be rewritten. Note that a good abstract should contain aim, methods, findings and recommendations
- Introduction - You should begin with the problem (motivation), the gap, then propose the research question and just after that say what they want to do to address that. Where is the gap? And you should make it clear why it is a gap? And if you claim there is a gap, then try to build a case for the gap.
- You should extend the literature review with application of heuristic algorithms in hub location and routing problems and discuss them to show gap. I suggest to the authors to read and discuss below interesting papers:
Das, M., Roy, A., Maity, S., Kar, S., & Sengupta, S. . (2022). Solving fuzzy dynamic ship routing and scheduling problem through new genetic algorithm. Decision Making: Applications in Management and Engineering, 5(2), 329–361.;
Mzili, T., Mzili,I., Riffi, M.E., Pamucar, D., Simic, V., Kurdi, M. (2023). A novel discrete rat swarm optimization algorithm for the quadratic assignment problem. Facta Universitatis-Series Mechanical engineering, 21(3), 529-552;
Mzili, I., Mzili, T., & Riffi , M. E. (2023). Efficient routing optimization with discrete penguins search algorithm for MTSP. Decision Making: Applications in Management and Engineering, 6(1), 730–743.;
- The conclusion section seems to rush to the end. The authors will have to demonstrate the impact and insights of the research. The authors need to rewrite the entire conclusion section with focus on both impact and insights of the manuscript. Clearly state your unique research contributions in the conclusion section. No bullets should be used in your conclusion section. Provide some future directions.

Experimental design

- Why do we need SA algorithm? Why not other heuristic algorithms like GA, hill-climbing algorithm, ABC, Grey Wolf etc.? Discuss advantages of these algorithms. What are advantages of SA over the mentioned algorithms?

Validity of the findings

- Provide more detail discussion on the results. A discussion section would allow you to come back to your research question and address the gap to the available literature you have defined. I also suggest to provide some correlation discussion with the results obtained based on other relevant algorithms in literature.
- Provide better validation section with comparisons with the existing models. Add more discussion on the results and discuss advantages and limitations.

Additional comments

- Addressing your research limitations could enhance the credibility, applicability, and impact of your research. It is important to note that limitations in a research paper do not necessarily imply negative aspects but rather areas that offer opportunities for further refinement and improvement. Address any potential drawbacks or constraints and how they were managed or could be improved in future iterations.

·

Basic reporting

no comment

Experimental design

the author constantly mixes heuristic and metaheuristic classes, which are not the same thing. When reading the article, it is very difficult to focus on the article's objective and value-added, as each time the author starts with an idea and continues to discuss someone else's idea without completing the first one. The contributions are not very clear; there are too many ideas. What we would like to see is why the SA method was chosen, what its role is, and why SA and not another method. The SA method is an iterative method that improves an initial solution by disturbing the initial solution; it is therefore an improvement method. However, the strong point of this method is the inclusion of an alert parameter linked to a problem, which allows accepting another solution after a certain time, which is positive and negative at the same time because it allows surpassing local optima but accepting solutions that do not make sense at certain times, which makes this method outdated and less used nowadays. The second remark is that the problem you are dealing with is a difficult, discrete problem, hence the need to see the transition from continuous to discrete. The method is designed the first time for continuous problems. However, to solve a discrete problem, it is necessary to show the transition, for example, how the problem is defined, how to define the solutions, the changes in solutions if it involves permutations of order, insertions... In all the tables, we saw that you compare your approach with metaheuristics; here you must mention the metaheuristics with which you are trying to compare. On the other hand, if you try to compare a class of algorithms with an entire class (metaheuristics), it is not logical and fair. The last remark is that to judge or compare two algorithms, it is necessary to describe the development environment, although it is impossible to have the same environment as others, but at least they will be more or less close to each other. And to validate the results, it is also necessary to base them on statistical tests such as Wilcoxon, Friedman, or ANOVA. I also invite you to check and confirm the times in all the tables because we have sometimes seen that the algorithm performed 20,706,028 iterations while it only took 0.0015 seconds. Overall, the article presents an interesting method, and the problem it addresses is difficult to solve and requires quality work.

Validity of the findings

no comment

Additional comments

Some articles to see:

http://casopisi.junis.ni.ac.rs/index.php/FUMechEng/article/view/11904

Part 5 Using PeSOA to Solve the MTSP: https://dmame-journal.org/index.php/dmame/article/view/731/154

https://linkinghub.elsevier.com/retrieve/pii/S111086651730066X

---

## Round 0.2 · accepted · Accept

All the reviewers' comments have been addressed carefully and sufficiently, the revisions are rational from my point of view, I think the current version of the paper can be accepted.

Reviewer 1 ·

Basic reporting

no comment

Experimental design

no comment

Validity of the findings

no comment

Additional comments

The authors have suitably revised the manuscript. It is recommended for publishing as it is.